# Prevalence of *Leishmania RNA virus* in *Leishmania* parasites in patients with tegumentary leishmaniasis: A systematic review and meta-analysis

**Endalew Yizengaw Shita**[1,2]*, **Endalkachew Nibret Semegn**[2,3], **Gizachew Yismaw Wubetu**[4], **Abaineh Munshea Abitew**[2,3], **Bizuayehu Gashaw Andualem**[3], **Mekuanint Geta Alemneh**[5,6]

**1** Department of Medical Laboratory Science, College of Medicine and Health science, Bahir Dar University, Bahir Dar, Ethiopia, **2** Institute of Biotechnology, Bahir Dar University, Bahir Dar, Ethiopia, **3** Department of Biology, College of Science, Bahir Dar University, Bahir Dar, Ethiopia, **4** Amhara Public Health Institute, Bahir Dar, Bahir Dar, Ethiopia, **5** Department of Medical Microbiology, School of Biomedical and Laboratory Sciences, College of Medicine and Health Sciences, University of Gondar, Gondar, Ethiopia, **6** Center for innovative Drug development and Therapeutic trials for Africa (CDT-Africa), College of Health Science, Addis Ababa University, Addis Ababa, Ethiopia

* endalew02@gmail.com

**Data Availability Statement:** All relevant data are within the manuscript and its Supporting Information files.

## Abstract

### Background

Cutaneous leishmaniasis is caused by different protozoan parasites of the genus *Leishmania*. *Leishmania RNA virus (LRV)* was identified as the first *Leishmania* infecting virus in 1998. Different studies showed the presence and role of the *LRV* in *Leishmania* parasites causing cutaneous leishmaniasis (CL). However, there is limited data on the pooled prevalence of *LRV* in *Leishmania* parasites causing CL. Therefore, the aim of this systematic review and meta-analysis was to determine the pooled prevalence of *LRV* in *Leishmania* parasite isolates and/or lesion biopsies in patients with CL from the available literature globally.

### Methodology

We retrieved the studies from different electronic databases. The studies were screened and identified based on the inclusion and exclusion criteria. We excluded studies exclusively done in experimental animals and *in vitro* studies. The review was conducted in line with PRISMA guidelines. The meta-analysis was performed with Stata software version 14 with metan command. The forest plot with random-effect model was used to estimate the pooled prevalence with 95% confidence interval. Inverse variance index ($I^2$) was used to assess the heterogeneity among the included articles.

### Principal findings

A total of 1215 samples from 25 studies were included. Of these, 40.1% (487/1215) were positive for *LRV*. The overall pooled prevalence of *LRV* globally was 37.22% (95% CI:

**Funding:** The author(s) received no specific funding for this work.

**Competing interests:** The authors have declared that no competing interests exist.

27.54% - 46.90%). The pooled prevalence of *LRV* in the New World (NW) and Old World (OW) regions was 34.18% and 45.77%, respectively. *Leishmania guyanensis*, *L. braziliensis*, *L. major*, and *L. tropica* were the most studied species for the detection of *LRV*. The prevalence of *LRV* from *Leishmania* isolates and lesion biopsies was 42.9% (349/813) and 34.3% (138/402), respectively.

## Conclusion

This systematic study revealed that there is high prevalence of *LRV* in *Leishmania* parasites isolated from patients with CL. More comprehensive studies would be required to investigate the presence of the *LRV* in other *Leishmania* species such as *L. aethiopica* to fully understand the role of *LRV* in different clinical manifestations and disease pathology presented in CL patients.

### Author summary

Cutaneous leishmaniasis (CL) is among the most neglected tropical diseases affecting a significant proportion of the world population, impacting mostly on the poorest communities. CL can present as localized, mucocutaneous or diffuse cutaneous. There are various reports on the prevalence of *Leishmania RNA virus* (*LRV*) in *Leishmania* parasites and/or lesion biopsies in patients with CL from both the Old World and New World regions. However, there is scarcity of comprehensive knowledge on the pooled prevalence of the virus in *Leishmania* parasites causing human CL. We searched different electronic databases and Google Scholar for published articles aimed to determine the presence of *LRV* in *Leishmania* parasites and/or lesion biopsies in patients with CL globally. The retrieved articles were screened according to the inclusion and exclusion parameters. Data was extracted based on the PRISMA guideline by reviewing the selected articles. Systematic review and meta-analysis would be one way to improve the level of evidence by providing pooled prevalence of the individual single studies regarding the presence of *LRV* in *Leishmania* parasites and/or lesion biopsies. This systematic review and meta-analysis revealed that a large proportion of the *Leishmania* parasites causing human CL harbour the endosymbiotic *LRV*. Further studies are needed to better understand the role of the virus in the clinical presentation of CL, as a potential target of treatment and vaccine development especially in Old World *Leishmania* parasites like *L. aethiopica*.

## 1 Introduction

The different clinical manifestations of tegumentary leishmaniasis comprise localized (LCL) mucocutaneous (MCL) and diffuse cutaneous leishmaniasis (DCL); they are neglected tropical diseases affecting millions of people worldwide. *Leishmania* species probably co-evolved with mammals [1] and have been identified in humans from at least 4 millennia before present [2]. About 431 million of the world population living in CL endemic areas are at risk of CL with 0.7–1.2 million new cases annually [3,4]. The mortality and morbidity of CL is also increasing [5,6]. It impacts mostly on the poorest community [5]. Cutaneous leishmaniasis is caused by a protozoan parasite of the genus *Leishmania* [7,2]. It is transmitted through the bites of female phlebotomine sandfly vectors. The infected sand fly vectors harbouring the parasite inoculate

the metacyclic promastigote stage into the skin of the host during blood meal [8]. There are more than 20 heterogenous *Leishmania* species known to cause CL. Old World CL, which occurs in Africa, Asia, and Europe, is predominantly caused by *L. tropica*, *L. major*, and *L. aethiopica* species. New World CL occurs in America and is mainly caused by *L. braziliensis*, *L. mexicana* and *L. amazonensis* [9,2].

While the majority of infected individuals remain asymptomatic, CL occurs in various clinical forms in symptomatic patients [10]. The clinical manifestation extends from a self-limited localized lesion to chronic and disfiguring mucocutaneous and diffused cutaneous lesions. These different clinical manifestations depend on several factors: the causative parasite, the immune response mounted by the host, and the sandfly factors [11,12]. Presence of *Leishmania RNA virus* in the parasite has also been reported to play a role in defining the clinical form of leishmaniasis [13,14]. However, there is no consensus on a defining role of *LRV* in disease progression, severity, metastasis, and treatment failure. Valencia *et al.*, 2022 did not find clinical, parasitological or immunological evidence supporting the hypothesis that *LRV1* is a significant determinant in the pathobiology of American Tegumentary Leishmaniasis [15]. Localised cutaneous leishmaniasis (LCL) is a benign form characterized by a single or a small number of lesions mostly around the face or extremities. It causes skin damage such as small ulcers and usually resolves spontaneously over time though the healing process is very slow [12]. However, the healed lesions generally result in permanent scars [16]. Mucocutaneous leishmaniasis (MCL) is the most severe form which leads to severe disfiguration and removal of the nose, mouth and/or the pharynx unless early treatment is initiated [17]. The pathophysiology of the ulcerative destruction is poorly understood, and relapse is common even after treatment and clinical cure [18,19]. Diffused cutaneous leishmaniasis (DCL) is a rare form of CL characterized by multiple non-ulcerative nodular lesions with large numbers of parasites in the lesions [20]. New World CL rarely evolves to self-cure; the lesions in most of the cases do not self-heal and are chronic, progressive and non-responsive to most anti-leishmanial drugs [18, 21]. Relapse is common even after successful treatment of MCL patients [22] and MCL often develops in more susceptible individuals after clinically healed LCL [18]. The distinct clinical manifestations are associated with different species, but there is substantial variation within species and it is poorly understood why *Leishmania* parasites cause different clinical manifestations.

The role of endosymbiotic *LRV* has been documented in the development of different clinical manifestations and disease severity of *Leishmania* infection [23,24] and it has been reported that it leads to the development of severe and disfiguring MCL [21]. The *LRV* is a double-stranded RNA virus belonging to the *Totiviridae* family [24]. Its genome has two open reading frames that encode for the capsid protein and the RNA polymerase [25–28]. The virus particle is composed of a capsid protein of about 40 nm in diameter and its genome has 5,280 nucleotides [25].

Research has focused on investigating different viruses infecting the *Leishmania* parasite and other unicellular eukaryotes since the discovery of the first protozoan virus in *Entamoeba histolytica* in 1960 [26]. *Leishmania RNA* virus was identified in 1998 as the first virus infecting *Leishmania* parasites [29]. Various studies, most of which are in the New World, have been conducted on the prevalence and role of the *LRV* in *Leishmania* parasites. However, there is limited comprehensive data scientifically analysed. Therefore, the aim of this systematic review and meta-analysis was to determine pooled prevalence of *LRV* in *Leishmania* parasite isolates and/or lesions in patients with cutaneous leishmaniasis. In our study the term tegumentary comprises localized, mucocutaneous and diffuse cutaneous leishmaniasis.

## 2. Methods

### 2.1. Search strategy and eligibility criteria

**2.1. Search strategy and eligibility criteria.** We first searched the articles using keywords and Medical Subject Headings (MeSH) in the following database: PubMed, Web of Science, Scopus, and Google Scholar; using the following key words:

- Leishmania RNA virus AND "cutaneous leishmaniasis";

- Leishmania RNA virus AND "localized cutaneous leishmaniasis"

- Leishmania RNA virus AND "mucocutaneous leishmaniasis";

- Leishmania RNA virus AND "diffuse leishmaniasis";

- Leishmania RNA virus AND "cutaneous leishmaniasis" AND "prevalence";

- Leishmania RNA virus AND "localized cutaneous leishmaniasis" AND prevalence"

- Leishmania RNA virus AND "mucocutaneous leishmaniasis" AND "prevalence";

- Leishmania RNA virus AND "diffuse leishmaniasis" AND "prevalence";

Only peer-reviewed original articles published in English were searched and no restriction was set by publication date, sample size, parasite species, methods used and study settings during article searching. We also had searched from the reference lists of all searched articles to further search similar studies and references.

### 2.2. Eligibility criteria and study selection

We included articles reporting primary data that aimed to determine the presence of *LRV* in *Leishmania* parasite isolates and/or lesion biopsies from patients with CL and MCL. We excluded studies exclusively done in experimental animals and *in vitro* studies. Review articles, non-CL based studies, and prevalence studies of other forms of leishmaniasis than CL were excluded. The review was conducted in line with Preferred Reporting Items for Systematic Reviews and Meta-analyses (PRISMA) guidelines (S1 Table) [30].

The searched articles were selected in a two-step process. First, we read each title of study, downloaded, and checked whether it was related with the review question. If it was found to be relevant, the abstracts were read and then the whole paper read, and all necessary data were extracted. The articles were independently reviewed by two investigators (EY, EN). In the second stage, the selected full-text articles were again reviewed for eligibility through detailed reading. The reasons for exclusion (studies focused on experimental model, method evaluation, immunology, case reports, non-CL based studies and studies using previously isolated parasites) were registered and reported when full-text articles were excluded. Finally, disagreements between the two investigators were resolved by discussions with all authors of this study until consensus was reached.

### 2.3. Data extraction and quality assessment

A data extraction form was prepared and used to extract the basic qualitative and quantitative data from each included article (S1 Data). Data from the included articles were extracted in parallel by two authors (EY, EN) independently and checked with each other when they had finished. The data included basic information from retrieved articles (the first author and year of publication), geographical region, method used to detect *LRV*, number and type of *Leishmania* species tested, and type of identified *Leishmania RNA virus*. The clinical manifestation

of the patients and the sample type used for detection of *LRV* were also extracted. The extracted data was entered into Microsoft Excel sheets. The quality of included studies was assessed by Joana Brigg's Institute (JBI) critical appraisal checklist for studies reporting prevalence data [31]. Articles of low quality that did not meet the JBI critical appraisal checklist were excluded.

### 2.4. Data analysis

The basic information of the included articles was summarized and presented in tables. The meta-analysis was done with Stata software (version 14, STATA Corp College Station, TX) with metan command. The extracted data was imported into the Stata software. The forest plot was used to estimate the combined prevalence and the effect of each study with their respective 95% confidence interval (CI).

Inverse variance index ($I^2$) was used to assess the extent of heterogeneity among the included articles in this systematic review and meta-analysis. The $I^2$ value ranges from 0 to 100%. $I^2$ values above 50%, between 25–50% and below 25% were considered as high, medium and low heterogeneity respectively [32]. P-value <0.05 was considered to conclude the presence of heterogeneity among the studies not by chance. Due to the presence of high heterogeneity about the prevalence among the included studies, we used random effect model at 95% CI to determine the pooled prevalence. We conducted subgroup analysis and meta-regression analysis was used to assess the possible causes for the observed heterogeneity. Publication bias across the included studies was assessed with Egger's funnel plot symmetry (qualitatively).

## 3. Results

### 3.1. Search results and the selection process

Initially, we retrieved a total of 3176 published articles from the preliminary searching of the English language electronic databases and manual searching. 206 duplicate articles were then removed from the total retrieved articles. After removing the duplicates, we screened the titles and abstracts of 2970 articles, and excluded 2882 articles (Fig 1). The full-text articles of the remaining 108 were assessed for eligibility. A further 83 articles were excluded after a detailed assessment considering all the inclusion and exclusion criteria: studies focused on experimental models, method evaluation, immunology, case reports, non-CL based studies and studies using previously isolated *Leishmania* parasites. The exclusion criteria are detailed in Fig 1.

Finally, 25 articles were found to be eligible and included in the systematic review and meta-analysis. The searched articles were identified following the preferred reporting items for systematic reviews and meta-analyses (PRIMSA) diagram (Fig 1).

### 3.2. Description of the included studies

The retrieved studies included in this systematic review and meta-analysis were published between 1998 and January 2022 with 36% published in 2019–2022. Most of the included studies (72%, 18/25) were conducted in New World regions. The presence of endosymbiotic *LRV* was detected in different *Leishmania* species causing CL. *Leishmania braziliensis*, *L. guyanensis*, *L. major*, *L. aethiopica*, *L. tropica*, *L. amazonensis* and *L. panamensis* were *Leishmania* species used in most of the studies. Different *Leishmania* species were diagnosed, isolated and used for the detection of *LRV* in some of the included studies.

Detection of the *LRV* was performed on *Leishmania* isolates and/or lesion biopsies. Regarding the diagnostic methods in the included articles, the presence of the virus was detected from parasite isolates and/or lesion biopsies by the polymerase chain reaction (PCR) in most of the

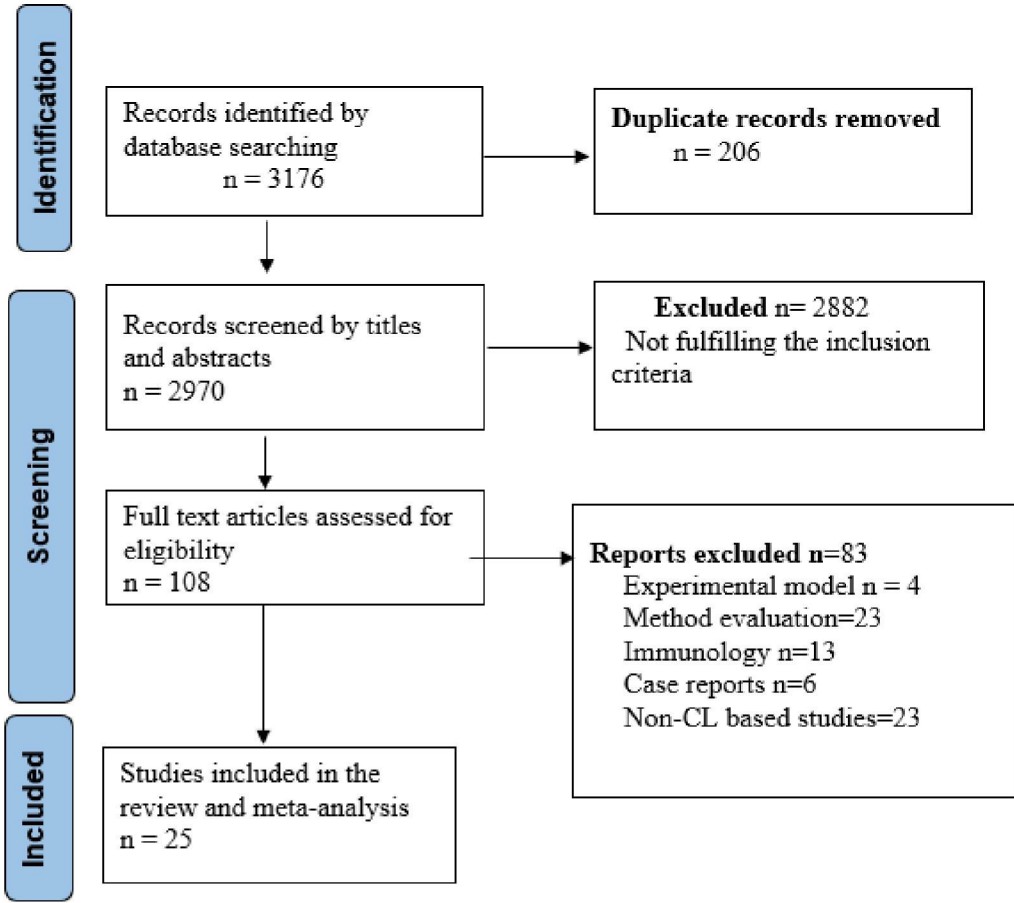

**Fig 1. PRISMA flow diagram describing the strategy for article selection for the prevalence of *LRV* in *Leishmania* parasites in patients with cutaneous leishmaniasis, 2021.**

studies. Electrophoresis, immunofluorescence microscopic techniques, and sequencing technologies were also used to detect *LRV* in some of the studies. Some of the included studies used different methods to detect *LRV*. *LRV1* was the type of *LRV* reported in most of the studies in New World regions and *LRV2* was reported from the Old World studies. Few studies reported both *LRV1* and *LRV2* in a single study. The *LRV* detected in Ethiopia from *L. aethiopica* was reported as "*LRV-Lae*". On the other hand, all the included articles were published as original articles except the study conducted by Valencia *et al*., 2014 [33] which was published as a conference abstract. The profiles of included articles are summarized in Table 1.

### 3.3. Synthesis of results

The forest plot was used to estimate the pooled prevalence with 95% confidence interval (CI). The heterogeneity among the studies included in this systematic review and meta-analysis was evaluated using inverse variance index ($I^2$). There was high heterogeneity among the included studies ($I^2$ = 92.6%), p = 0.000. Due to the presence of high heterogeneity about the prevalence among studies, we used random effect model at 95% CI for the analysis of the pooled prevalence of *LRV* in *Leishmania* isolates and/or lesions from patients with CL. Subgroup analysis and meta-regression analysis were used to assess the possible causes for the observed heterogeneity.

**Table 1. Overview of the studies describing the presence of *LRV* in *Leishmania* parasites and/or lesions in patients with CL, 2021.**

| Author, Year (ref) | *Leishmania* species | Methods | Sample type | Parasites no | Pos. *LRV* (%) | Type of *LRV* | Region |
|---|---|---|---|---|---|---|---|
| Tarr *et al.*, 1988 [29] | *L. braziliensis, L. guyanensis* | Electrophoresis | Isolates | 12 | 2 (16.7) | LRV2 | NW |
| Widmer *et al.*, 1989 [35] | *L. guyanensis* | Electrophoresis | Isolates | 10 | 2 (20) | LRV1 | NW |
| Scheffter *et al.*, 1995 [27] | *L. major* | Electrophoresis | Isolates | 14 | 6 (42.8) | LRV2-1 | OW |
| Zangger *et al.*, 2014 [46] | *L.aethiopica* | IFM, dot plot, PCR | Isolates | 8 | 4 (50) | LRV-Lae | OW |
| Saberi *et al.*, 2020 [45] | *L. major, L. tropica* | PCR | Isolates | 85 | 59 (69.4) | LRV2 | OW |
| Abtahi *et al.*, 2020 [39] | *L. major* | PCR | Isolates | 30 | 9 (30) | LRV2 | OW |
| de Carvalho *et al.*, 2019 [42] | *L. guyanensis* | PCR | Isolates | 49 | 19 (38.8) | LRV1 | NW |
| Cantanhêde *et al.*, 2015 [13] | *L. braziliensis, L. amazonensis, L. guyanensis* | PCR, Sequencing | Lesion | 141 | 61 (39) | LRV1 | NW |
| Nalçacı *et al.*, 2019 [47] | *L. tropica, L. major* | PCR, Sequencing | Isolates | 25 | 8 (32) | LRV2 | OW |
| Kleschenko *et al.*, 2019 [48] | *L. major* | Sequencing | Isolates | 3 | 2 (66.7) | LRV2 | OW |
| Kariyawasam *et al.*, 2019 [37] | *L. braziliensis, L. guyanensis, L. panamensis* | PCR | Isolates | 8 | 4 (50) | LNV1 | NW |
| Kariyawasam *et al.*, 2020 [55] | *L. braziliensis* | PCR | Isolates | 78 | 21 (26.9) | LNV1 | NW |
| Ito *et al.*, 2015 [49] | *L. braziliensis, L. guyanensis* | PCR | Isolates | 37 | 26 (70.3) | LNV1 | NW |
| Bourreau *et al.*, 2016 [41] | *L. guyanensis* | PCR | Lesion | 75 | 44 (58.7) | LNV1 | NW |
| Adaui *et al.*, 2016 [40] | *L. braziliensis* | PCR, Sequencing | Isolates | 97 | 32 (33) | LRV1 | NW |
| Ginouvès *et al.*, 2016 [36] | *L. guyanensis, L. braziliensis* | PCR | Isolates | 129 | 96 (74.4) | LRV1 | NW |
| Hartley *et al.*, 2016 [56] | *L. guyanensis* | PCR | Isolates | 78 | 30 (38.5) | LRV1 | OW |
| Ogg *et al.*, 2003 [57] | *L. braziliensis* | PCR | Lesion | 36 | 12 (25.5) | LRV1 | NW |
| Parra-Muñoz *et al.*, 2021 [54] | *L. braziliensis* | PCR | Isolates & Lesion | 47 | 15 (41.7) | LRV1 | NW |
| Kariyawasam *et al.*, 2017 [50] | *L. braziliensis, L. panamensis* | PCR | Isolates | 5 | 3 (60) | LRV1 | NW |
| Pereira *et al.*, 2013 [52] | *L. guyanensis* | PCR | Lesion | 48 | 2 (4.2) | LRV1 | NW |
| Saiz *et al.*, 1998 [53] | *Leishmania* species | PCR | Lesion | 11 | 2 (18.2) | LRV1-1, LRV1-4 | NW |
| Guilbride *et al.*, 1992 [34] | *L. braziliensis, L. guyanensis* | Electrophoresis | Isolates | 71 | 12 (16.9) | LRV1 | NW |
| Valencia *et al.*, 2014 [33] | *L. braziliensis, L. guyanensis, L. peruviana* | PCR | Isolates | 56 | 6 (10.7) | LRV1 | NW |
| Valencia *et al.*, 2022 [15] | *L. braziliensis, L. peruviana, L. guyanensis* | PCR | Lesion | 56 | 10 (17.8) | LRV1 | NW |

Abbreviations: IFM: Immunofluorescence Microscopy; L: *Leishmania*; PCR: Polymerase Chain Reaction; OW: Old World; NW: the New World

The data retrieved from the included articles was analysed by meta-analysis to determine the pooled prevalence of *LRV* in the *Leishmania* isolates and /or lesions of patients with CL. A total of 1215 samples (promastigote isolates and lesions) were tested for the presence of the virus, of which 40.1% (487/1215) were positive for *LRV*. Using random effect model analysis, the overall pooled prevalence was 37.22% with 95% CI (27.54–46.90%) (Fig 2).

**3.3.1. Publication bias assessment.** Egger's funnel plot symmetry test for small study effects was used to check the publication bias among the included articles. The results showed that publication bias was not significant ($p = 0.213$) in the included studies (Fig 3)

We included studies done in New World and Old World regions. The prevalence of *LRV* was 34.18% and 45.77% in the New World and Old World regions, respectively. Regarding the clinical manifestations of CL, 40.3% (381/946) of samples isolated from patients with localized

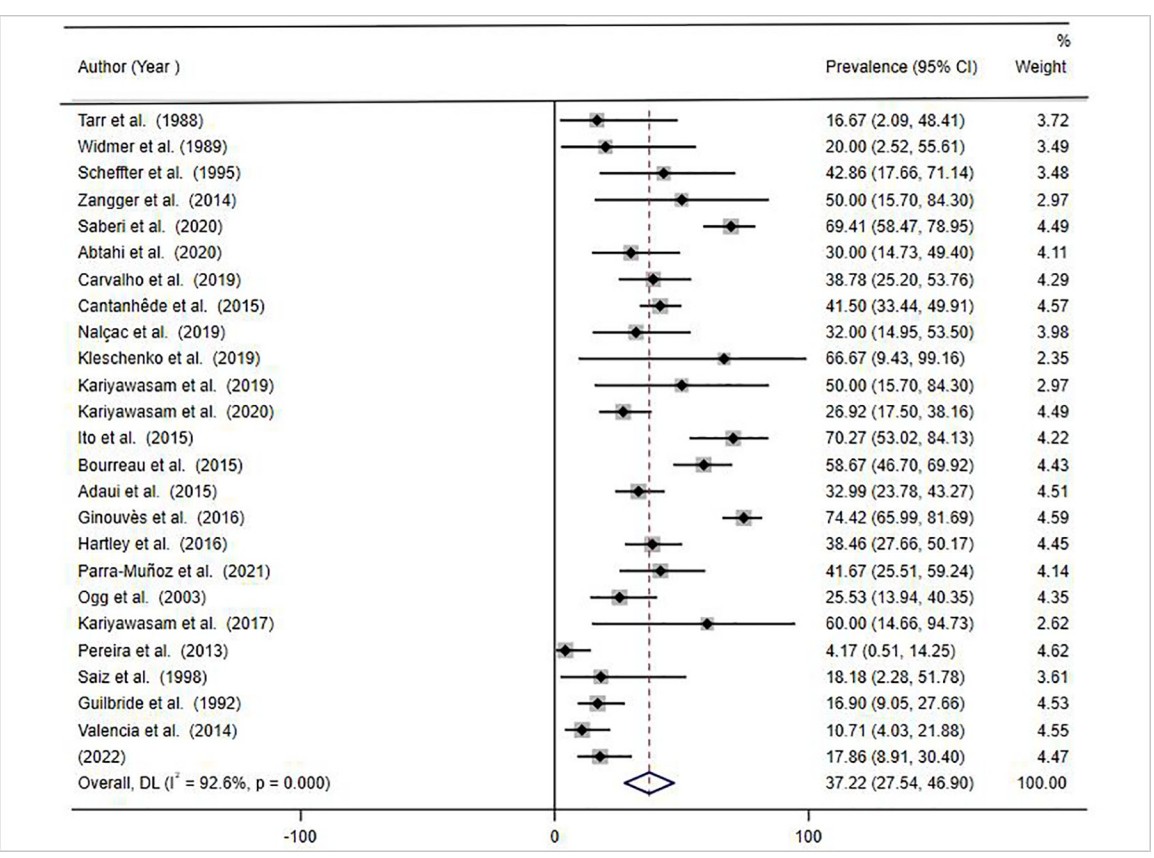

**Fig 2. Forest plot diagram of included studies depicting the pooled and weighted prevalence of *LRV* in *Leishmania* isolates and/or lesions from CL patients, 2021.**

cutaneous leishmaniasis were positive for *LRV*. *Leishmania guyanensis*, *L. braziliensis*, *L. major*, and *L. tropica* were the most common *Leishmania* species included in studies for the detection of *LRV*. Zangger *et al.*, 2014 used *L. aethiopica* and 50% of samples were positive for *LRV*. Moreover, the prevalence of *LRV* from *Leishmania* isolates and lesion biopsies was 42.9% (349/813) and 34.3% (138/402), respectively (Table 2).

## 4. Discussion

This systematic review and meta-analysis aimed to determine the comprehensive prevalence of *LRV* in different *Leishmania* species causing human CL. The *Leishmania* parasite has been known to harbour the endosymbiotic *LRV* since 1988 [28] and the early 1990s [34,35]. Different studies have been conducted to determine the status of the *LRV* in different *Leishmania* species since the first evidence in *L. guyanensis* and *L. braziliensis* in 1988 by Tarr *et al.*, 1988 [36]. The pooled *LRV* prevalence in parasite isolates and/or lesion biopsies from patients with cutaneous leishmaniasis was 37.22%, 95% CI (27.54%-46.90%). This reveals that a significant proportion of the *Leishmania* parasites causing human CL harbour the endosymbiotic *LRV*. This might be one of the reasons for the increasing burden, different clinical presentation and severity of CL [13,37]. Evidence shows that the severe forms of CL including disseminated and diffused leishmaniasis are emerging and increasing due to infection with different *L. (viannia) braziliensis* parasites [38]. The presence of *LRV* in the *Leishmania* parasite was also associated with failure to respond to antimonial treatment and to relapse of the symptomatic disease [39–

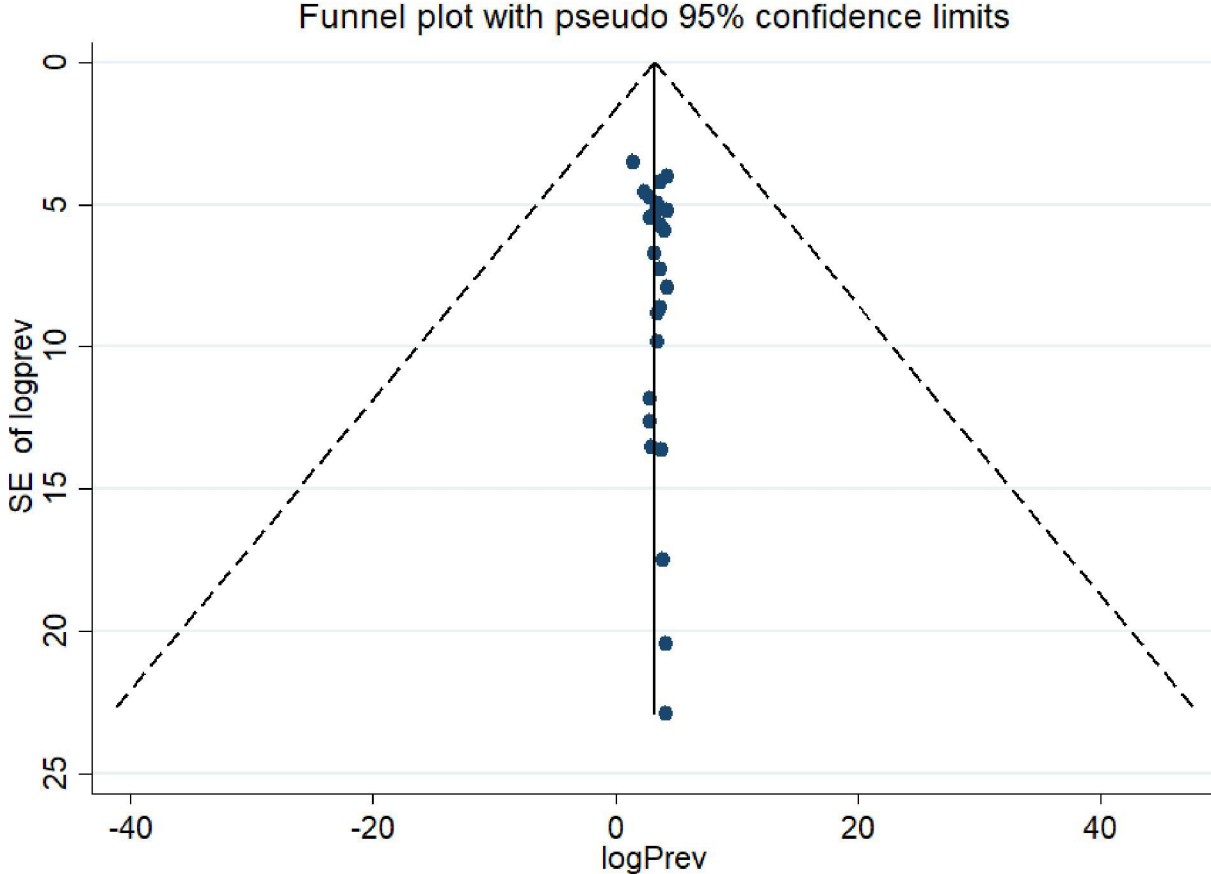

**Fig 3. Egger's funnel plot indicates absence of publication bias across the included studies, 2021.**

41]. However, the role of *LRV* on the pathobiology of cutaneous leishmaniasis is still unclear and contradictory data showing lack of association of *LRV* positivity and treatment failure have been reported [15]. It has been indicated that in the presence of the *LRV* in the *Leishmania* parasite, the immune response favours parasite survival and results in severe disease. The dsRNA of the virus acts as potent innate immunogen recognized via toll like receptor-3 (TLR-3) [42]. According to the findings from studies conducted in *L. guyanensis* parasites, there is an association between *LRV1* and antimonial treatment failure [40,41]. This might suggest that *LRVs* can be considered as a potential target for treatment and prevention. There is evidence that *LRV1* can be found in exosome vesicles within the *Leishmania* parasite [43]. This would explain the role of the *LRV* virus in the development of severe leishmaniasis as the vesicle helps the virus to disseminate further. However, recent reports have shown that *LRV* has no role in the disease severity, determination of clinical forms, and treatment failure [15].

Articles included in this systematic review and meta-analysis used various methods to detect the presence of *LRV* in *Leishmania* parasite isolates and/or lesions of patients with CL. Most of the studies used PCR techniques of different types, summarised as PCR for simplicity of analysis. Some of the studies used more than one detection method to increase the sensitivity. The type of detection method was among the variables for the heterogeneity of the studies (Fig 4 below).

Currently, *LRV* has been detected in *Leishmania* parasites isolated from both New World (in Central and South America regions) and Old World (Central, Western and Eastern

**Table 2. Summarized data extracted from the included studies for sub-group analysis in the meta-analysis.**

| Category | No. | Positive for *LRV* | Prevalence (%) |
|---|---|---|---|
| **Type of *LRV*** | | | |
| *LRV*1 | 1044 | 398 | 38.12 |
| *LRV*2 | 163 | 85 | 52.1 |
| *LRV-Lae* | 8 | 4 | 50.0 |
| Total | 1215 | 487 | 40.1 |
| **Clinical Form** | | | |
| CL | 946 | 381 | 40.3 |
| MCL | 237 | 99 | 41.8 |
| DCL | 32 | 7 | 21.9 |
| Total | 1159 | 477 | 41.1 |
| Metastatic | 269 | 106 | 39.4 |
| Non-metastatic | 946 | 381 | 40.3 |
| Total | 1215 | 487 | 40.1 |
| **Leishmania Species** | | | |
| *Leishmania guyanensis* | 473 | 208 | 44.0 |
| *Leishmania braziliensis* | 518 | 179 | 34.5 |
| *Leishmania major* | 133 | 78 | 58.6 |
| *Leishmania aethiopica* | 8 | 4 | 50 |
| *Leishmania tropica* | 24 | 6 | 25 |
| *Leishmania* species | 56 | 12 | 21.4 |
| Total | 1215 | 487 | 40.1 |
| **Sample type** | | | |
| Isolates | 813 | 349 | 42.9 |
| Lesion | 402 | 138 | 34.3 |
| Total | 1215 | 487 | 40.1 |

Europe, Far and Middle East, and Africa regions) patients with CL. Most of the studies were focused on *L. guyanensis*, *L. braziliensis*, *L. panamensis* species of the New World region and reported *LRV1*. Recent evidence showed that *LRV2* has also been detected in *L. major* and *L. aethiopica*, *Leishmania* species of the Old World [44]. Sub-group analysis indicates that the prevalence of *LRV* in the Old World was higher (45.77%) than the New World (34.18%) (Fig 5). This might be because more samples per study were included in studies conducted in the Old World regions than the New World regions. The variation in the level of endemicity of *LRV* in the two geographic areas might also explain the difference. Saberi *et al.*, 2020 [45] tested 85 samples of *L. major* and *L. tropica* isolated from patients with CL from Iran. Out of these, 59 samples were *LRV2* positive. There is limited study in the Old World regions regarding *LRV* in *Leishmania* parasite isolates and/or samples from patients with CL. However, a study conducted in *L. aethiopica* parasites in Ethiopia reported the presence of *LRV2* in 5 out of 11 *L. aethiopica* isolates [46]. In addition, a study conducted in Turkey indicated 7 out of the 24 *L. tropica* and 3 out of the 3 *L. major* isolates were *LRV2* positive [47]. Moreover, *LRV2* was detected in two out of 3 *L. major* isolates from human patients with CL in south Uzbekistan [48].

*Leishmania RNA Virus* has been isolated from different *Leishmania* isolates (Fig 6). Higher *LRV* prevalence was detected in *Leishmania guyanensis* and *Leishmania braziliensis*. Similar findings have been reported by Saberi *et al*, 2019 [44]. They reviewed different studies aimed to detect *LRV* in *Leishmania* parasite including *Leishmania infantum*. They indicated that the

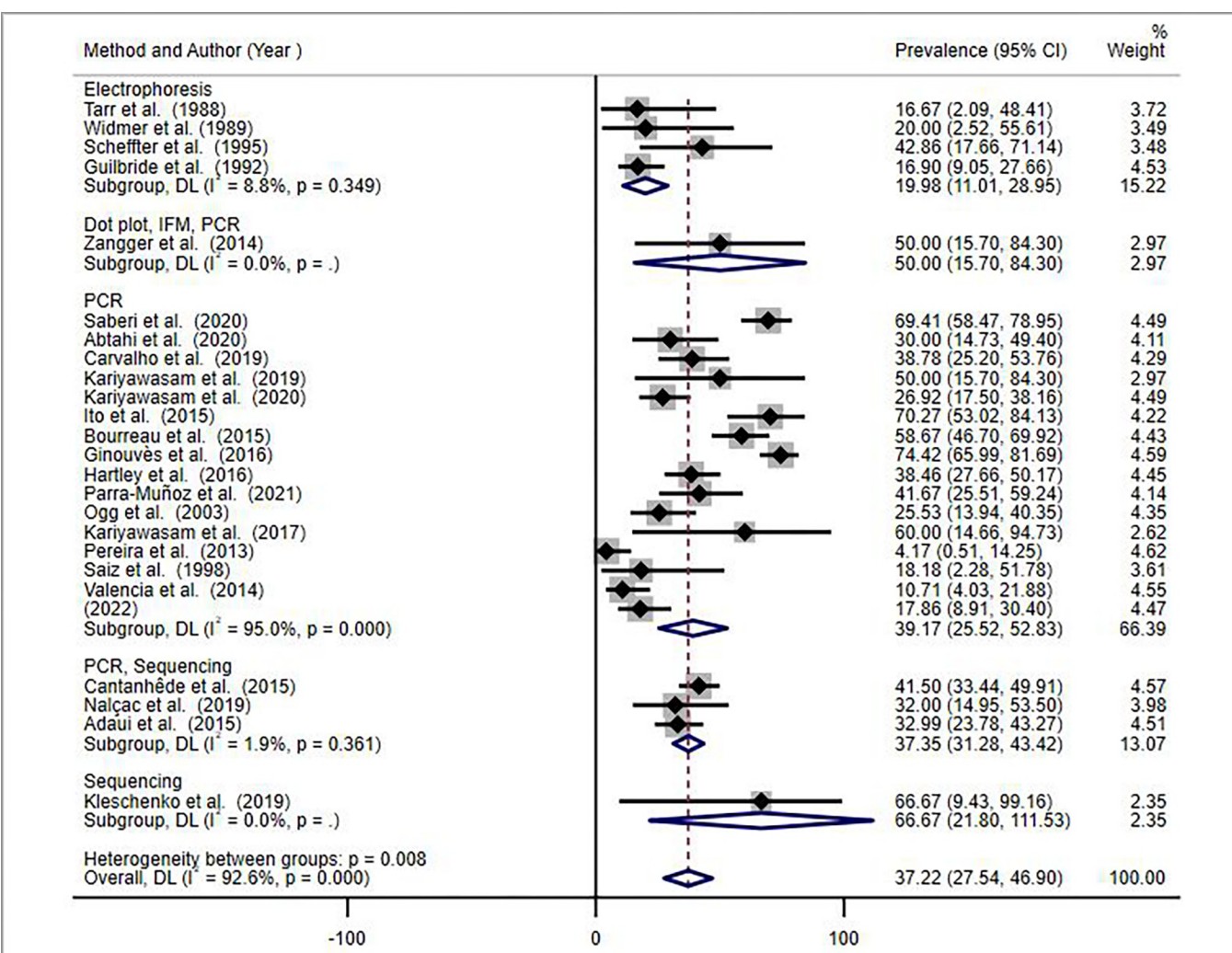

**Fig 4. Forest plot showing the pooled *LRV* prevalence estimate by detection method in *Leishmania* parasites and/or lesions of patients with CL, 2021.**

high prevalence of *LRV* among causative agents of New World *Leishmania* isolated from the metastatic clinical forms suggests potential association of *LRV* with metastatic clinical forms in New World endemic regions. However, Valencia *et al*, 2022 showed that the presence of LRV is not correlated with different clinical forms, treatment failure and disease severity [15]. Our study as well as other publications [15,44] illustrate that there is a gap of knowledge and more work need to be carried out to reach consensus on the role of LRV in CL.

Considering the clinical presentations, *LRV* has been reported from all clinical manifestations of CL with higher prevalence in metastatic form (MCL and DCL) than non-metastatic one (LCL). This is in line with different reports that have described that *LRV* results in disease severity and leads to the metastatic and disfiguring form of CL [33,47,48,49]. Ives *et al.*, 2011 showed that metastasizing parasites have a high *Leishmania RNA virus–1* (*LRV1*) burden that is recognized by the host Toll-like receptor 3 (TLR3) of immune cells to induce proinflammatory cytokines and chemokines [14]. This intense proinflammatory condition leads to tissue destruction and disfiguring [46,50]. Besides, the majority of the *LRVs* were detected in *L. braziliensis* (n = 518) and L. *guyanensis* (n = 473) parasites that are the causative agents for MCL

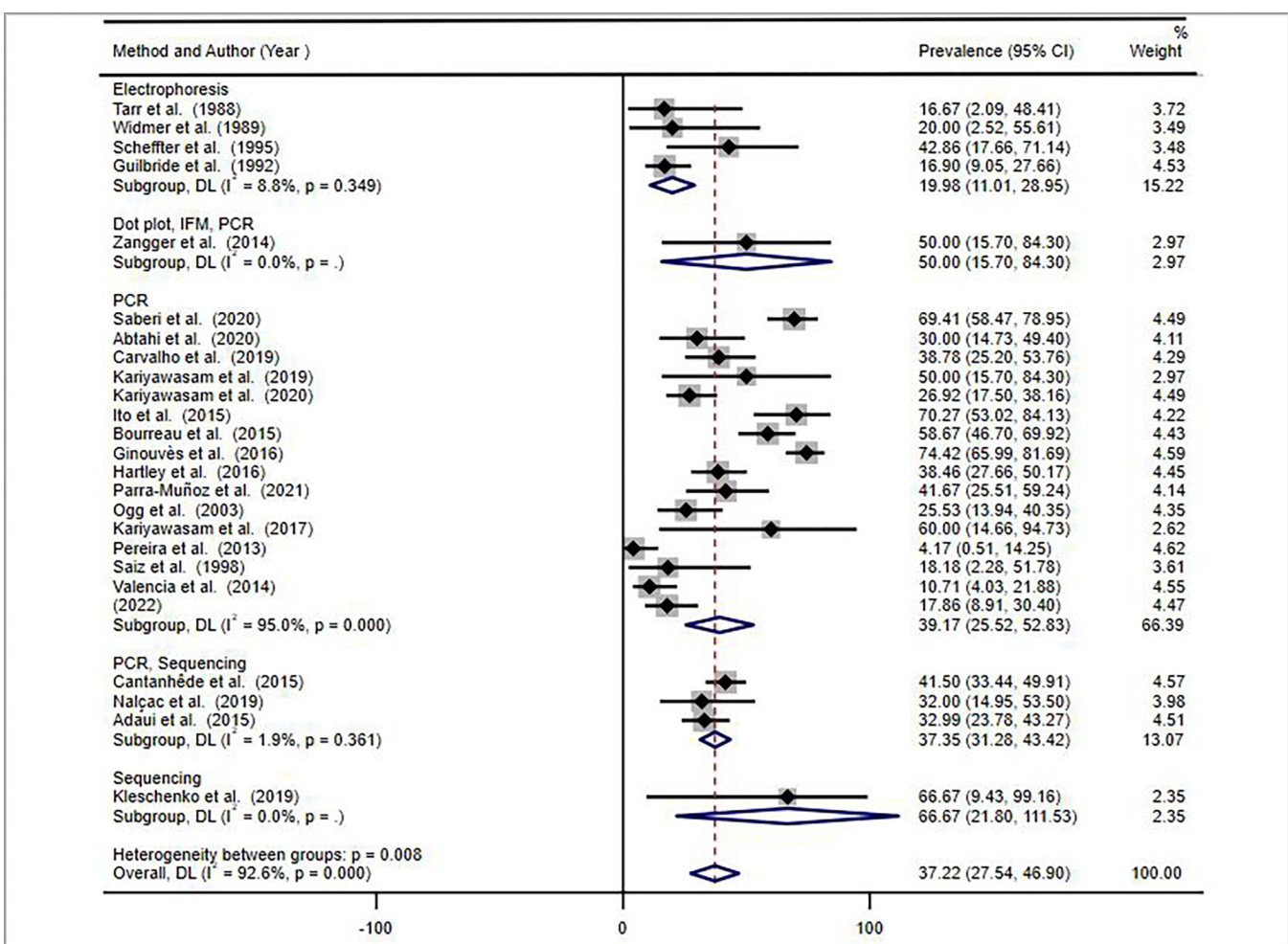

**Fig 5. Forest plot showing the pooled *LRV* prevalence estimate by geographical region in *Leishmania* parasites and/or lesions of patients with cutaneous leishmaniasis, 2021.**

[51]. On the other hand, it has been indicated that the presence of *LRV* might not be the only factor explaining the evolvement of the disease to severe forms [52].

It has also been described that there is variation in the intensity of the viral load and sequence divergence of *LRV* among different *Leishmania* parasites [33,46]. Different *Leishmania* parasites contain related viruses with distinct regions of sequence conservation and divergence in their genome. The presence of the virus and the intensity of the viral load can also be detected in lesion biopsies [53–57]. The detection of the *LRV* directly from the lesion biopsies would be much easier than detecting it from cultured parasites since it does not require parasite culturing and would help to run more samples in a short period of time in areas with poor laboratory setup. This might also avoid the effect of the culturing and processing procedures on the *LRV* status. The culturing environment of the *Leishmania* parasite may not be favourable for the endosymbiotic virus. Despite limited data available on the area to reach more reliable results, our study is relevant for the community since it shows the prevalence of LRV in isolates of different species of *Leishmania*.

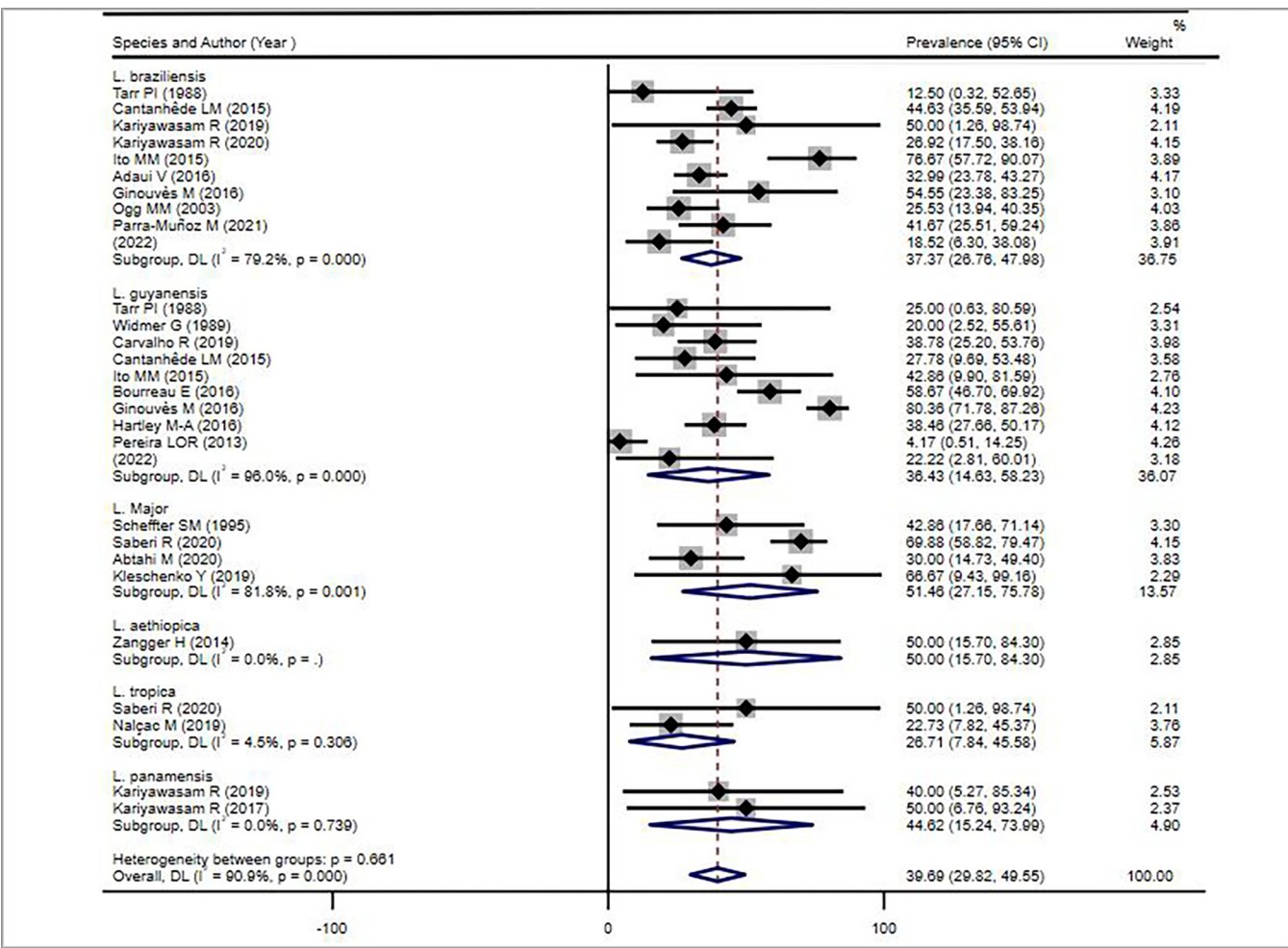

**Fig 6. Forest plot showing the pooled *LRV* prevalence estimate in different *Leishmania* species from patients with cutaneous leishmaniasis, 2021.**

## 5. Limitations

The main limitation of our study is the scarcity of published data on *LRV* in localized cutaneous, mucocutaneous and diffuse cutaneous leishmaniasis. The sample size used in most of the available data was limited too. There is not enough information on the role of *LRV* in the different clinical manifestation of cutaneous leishmaniasis. The mechanism by which the presence of *LRV* affects the disease pathogenesis, disease severity, treatment failure and relapse has not been identified and is not addressed in most of the included studies. We also did not show the impact of the viral load and sequence divergence of *LRV* among different *Leishmania* parasites on the disease outcome.

## 6. Conclusion

This systematic review and meta-analysis revealed that there is high prevalence of *LRV* in *Leishmania* isolates and/or lesion biopsies from patients with CL globally. Future studies using large numbers of isolates are needed to collect more information on the presence of the *LRV* in *Leishmania* parasites. Conducting experimental and clinical based studies will be key to better understand the role of the virus in clinical presentations of the disease, treatment failure and disease severity.

## Supporting information

**S1 Table. PRISMA 2009 checklist.**
(DOCX)

**S1 Data. Data extraction form and the whole data set.**
(XLS)

## Acknowledgments

We would like to thank Mr. Abebaw Bitew, a PhD fellow, University of Gondar and Dr. Mengistie Taye, Head of Biotechnology Research Institute, Bahir Dar University for their unreserved support throughout this work and Edward Cruz Cervera, Imperial College, London for critical reading of the manuscript

## Author Contributions

**Conceptualization:** Endalew Yizengaw Shita, Bizuayehu Gashaw Andualem, Mekuanint Geta Alemneh.

**Data curation:** Endalew Yizengaw Shita, Endalkachew Nibret Semegn, Abaineh Munshea Abitew.

**Formal analysis:** Endalew Yizengaw Shita, Endalkachew Nibret Semegn, Mekuanint Geta Alemneh.

**Methodology:** Endalew Yizengaw Shita, Endalkachew Nibret Semegn, Gizachew Yismaw Wubetu, Abaineh Munshea Abitew, Bizuayehu Gashaw Andualem.

**Software:** Endalew Yizengaw Shita, Endalkachew Nibret Semegn, Abaineh Munshea Abitew, Mekuanint Geta Alemneh.

**Validation:** Endalew Yizengaw Shita, Endalkachew Nibret Semegn, Gizachew Yismaw Wubetu, Abaineh Munshea Abitew, Bizuayehu Gashaw Andualem, Mekuanint Geta Alemneh.

**Writing – original draft:** Endalew Yizengaw Shita, Bizuayehu Gashaw Andualem, Mekuanint Geta Alemneh.

**Writing – review & editing:** Endalkachew Nibret Semegn, Gizachew Yismaw Wubetu, Abaineh Munshea Abitew.

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
