## [Decision Letter · Decision Letter 0]

6 Jan 2022

Dear Prof Shita

Thank you very much for submitting your manuscript "Prevalence of Leishmania RNA virus in Leishmania parasites in patients with cutaneous leishmaniasis: a systematic review and meta-analysis" for consideration at PLOS Neglected Tropical Diseases. As with all papers reviewed by the journal, your manuscript was reviewed by members of the editorial board and by several independent reviewers. In light of the reviews (below this email), we would like to invite the resubmission of a significantly-revised version that takes into account the reviewers' comments. 

The manuscript was revised by three different reviewers. I strongly suggest taking into account the comments of the second and third reviewers. Despite little evidence in the literature, the data presented are of interest, but I suggest that there is a better explanation regarding the presence of LRV and the development of clinical forms of tegumentary leishmaniasis. The review cannot be based only on cutaneous leishmaniasis, since there are data associating mucosal leishmaniasis and the presence of LRV and, therefore, the word cutaneous should be replaced by tegumentary in the title of the manuscript.

We cannot make any decision about publication until we have seen the revised manuscript and your response to the reviewers' comments. Your revised manuscript is also likely to be sent to reviewers for further evaluation.

Sincerely,

José Angelo Lauletta Lindoso

Guest Editor

Fabiano Oliveira

Deputy Editor

The manuscript was revised by three different reviewers. I strongly suggest taking into account the comments of the second and third reviewers. Despite little evidence in the literature, the data presented are of interest, but I suggest that there is a better explanation regarding the presence of LRV and the development of clinical forms of tegumentary leishmaniasis. The review cannot be based only on cutaneous leishmaniasis, since there are data associating mucosal leishmaniasis and the presence of LRV and, therefore, the word cutaneous should be replaced by tegumentary in the title of the manuscript.

Reviewer's Responses to Questions

**Key Review Criteria Required for Acceptance?**

**Methods**

-Are the objectives of the study clearly articulated with a clear testable hypothesis stated?

-Is the study design appropriate to address the stated objectives?

-Is the population clearly described and appropriate for the hypothesis being tested?

-Is the sample size sufficient to ensure adequate power to address the hypothesis being tested?

-Were correct statistical analysis used to support conclusions?

-Are there concerns about ethical or regulatory requirements being met?

Reviewer #1: see general comments below

Reviewer #2: -Are the objectives of the study clearly articulated with a clear testable hypothesis stated? Answer: YES

-Is the study design appropriate to address the stated objectives?Answer: Yes, because the authors used the PRISMA guidelines

-Is the population clearly described and appropriate for the hypothesis being tested?Answer: No , because there are few published studies avialable to be analyzed 

-Is the sample size sufficient to ensure adequate power to address the hypothesis being tested?Answer: NO , because there are few published studies avialable to be analyzed

-Were correct statistical analysis used to support conclusions?Answer: YES

-Are there concerns about ethical or regulatory requirements being met?Answer: Yes

Reviewer #3: The aim of the study is clearly presented and the study design described thoroughly. 

It is a review and meta-analysis MS and the selection criteria for the articles included for analysis are clearly presented. 

In the introduction, some informations are not based in appropriate references such as case reports and visceral leishmaniasis paper (referring to asymptomatic cases). We notice that the authors are not aware that New World CL rarely evolves to self-cure; this information needs correction. 

Among the articles included some were based in small number of parasite isolates and it is not clear wether in these articles the isolates were randomly included or they had any bias for the inclusion in the study.

**Results**

-Does the analysis presented match the analysis plan?

-Are the results clearly and completely presented?

-Are the figures (Tables, Images) of sufficient quality for clarity?

Reviewer #1: see general comments below

Reviewer #2: -Does the analysis presented match the analysis plan? Answer: YES

-Are the results clearly and completely presented?Answer:NO, not because the data extraction form is missing 

-Are the figures (Tables, Images) of sufficient quality for clarity?Answer: No not because the prevalence could be separated by species

Reviewer #3: To estimate the prevalence of LRV, from the data presented in the Table 2, it seems that the presence of the virus does not relate to particular clinical presentation and the development of severe forms.

There is no other data that show correlation of the presence of the virus and worse disease development. It would be desirable the analysis of the data from articles showing high number of isolates where it would be possible to compare the development of the disease with parasites with and without the virus avoiding the confounding factors such as geographical area and Leishmania species diversity.

**Conclusions**

-Are the conclusions supported by the data presented?

-Are the limitations of analysis clearly described?

-Do the authors discuss how these data can be helpful to advance our understanding of the topic under study?

-Is public health relevance addressed?

Reviewer #1: see general comments below

Reviewer #2: -Are the conclusions supported by the data presented? Answer: no , Because some conclusions are not a consensus in the area

-Are the limitations of analysis clearly described?Answer:No , because it was not discussed that there is little bibliographical information available 

-Do the authors discuss how these data can be helpful to advance our understanding of the topic under study?Answer:YES but could better discuss

-Is public health relevance addressed?Answer:YES but could better discuss

Reviewer #3: Conclusions are not fully supported by the presented data. Limitations of the analysis are not clearly described. 

The relevance of the study for the public health is not clearly addressed.

**Editorial and Data Presentation Modifications?**

Reviewer #1: could do with a native English speaker improving English in places

Reviewer #2: Introduction line 5 not necessarily leishmaniasis is prevalent in the poorest communities . paragraph 2 line 5

there is no consensus that LV can define the clinical form.paragraph 4 line 8 the authors could clarify whether the isolates were obtained from patients.

methods: 

2.1 it was not clear the search strategy, it was not clear whether the keywords were used separately, together or concatenated.

2.2 why? all in vitro and animal model studies were eliminated, as these studies can be done with patient isolates.paragraph 2 lines 6 and 7 authors could provide the exclusion criteria made at this stage of the analysis

2.3 line 1 authors could provide the data extraction form and citeria.line 9 authors could make it clear if the "checklist-JBI" were used as exclusion criteria.

3.1line 4 the authors could indicate that these results are described in figure 1. line 5 the authors can describe how the 83 studies from the last 107 were excluded and indicate that the exclusion criteria is described in figure 1.

3.3 in line 4 what does "p=0.00." means? .

Discussion lines 9-10 there is no consensus that LV defines the clinical form of cutaneous leishmaniasis.lines 13-14 there is no consensus that LV causes therapeutic failure.line 15 separate leishmania from parasite.

Figure 4 was not shown in the results.

page 18-19 line 2 is not a consensus

Reviewer #3: Language revison and edition would be desirable.

**Summary and General Comments**

Reviewer #1: Prevalence of Leishmania RNA virus in Leishmania parasites in patients with cutaneous leishmaniasis: a systematic review and meta-analysis.

This a useful and well performed study, and highlights areas relevant knowledge gaps and areas for future research.

My comments mostly relate to areas of text which could be clearer. 

Principal findings abstract: ‘total of 1159 samples from 24 studies were tested’ makes it sound as if you tested them. 

Author summary: ‘It needs further studies to well understand’ would read better as ‘It needs further studies to better understand…’

Your opening sentence is too vague ‘Cutaneous leishmaniasis (CL) is among neglected tropical diseases affecting millions of the world population since long years back [1, 2].’

I would suggest something like ‘Cutaneous leishmaniasis (CL) is a neglected tropical diseases affecting millions of people worldwide. Leishmania species probably co-evolved with mammals (Steveding et al 2017) and have been identified in humans from at least 4 millennia before present (Zink et al).’

multiplenon-ulcerative nodular lesionswith. should be : multiple non-ulcerative nodular lesions with lar

methods- can you be clearer about your search strategy. You write ‘combinations of search terms: “cutaneous leishmaniasis “AND “Leishmania RNA virus OR “leishmaniasis, cutaneous” OR mucocutaneous, prevalence.”

Was this combinations of search terms: “cutaneous leishmaniasis “AND (“Leishmania RNA virus OR “leishmaniasis, cutaneous” OR mucocutaneous, prevalence.”)?

Surely RNA virus was always in the search strategy? In which case I would have expectd Leishmania RNA virus AND (“cutaneous leishmaniasis “ “OR “leishmaniasis, cutaneous” OR mucocutaneous, prevalence.”)

Is this correct? Please clarify

‘The articles were independently reviewed by two investigators (EY, EN)’. What did you do if you disagreed?

Discussion: ‘The detection of the LRV from the lesion biopsies would be much easier and help to run more samples in short period of time in areas with poor laboratory setup.’ I don’t understand this sentence. Please clarify.

Reviewer #2: the authors can discuss the article titled "Global status of synchronizing Leishmania RNA virus in Leishmania parasites: A systematic review with meta‐analysis" because are similar and to discuss limitations of the present study, discuss some similar finding to, argue that more studies need to be carried out to reach more reliable results and at the same time to emphasize that there is a gap in this area of knowledge.

Reviewer #3: The MS is relevant to show the prevalence of LRN in the isolates of different species of Leishmania. However, the main concern that is the correlation of the presence of the virus with the disease development is not clearly presented based on the supporting data.

PLOS authors have the option to publish the peer review history of their article (what does this mean?). If published, this will include your full peer review and any attached files.

Reviewer #1: Yes: Dr Richard Weller

Reviewer #2: No

Reviewer #3: No
---

## [Decision Letter · Decision Letter 1]

22 Mar 2022

Dear Mr Shita,

Thank you very much for submitting your manuscript "Prevalence of Leishmania RNA virus in Leishmania parasites in patients with tegumentary leishmaniasis: a systematic review and meta-analysis" for consideration at PLOS Neglected Tropical Diseases. As with all papers reviewed by the journal, your manuscript was reviewed by members of the editorial board and by several independent reviewers. The reviewers appreciated the attention to an important topic. Based on the reviews, we are likely to accept this manuscript for publication, providing that you modify the manuscript according to the review recommendations. 

The manuscript was revised after some changes made by the authirs. The answers provided by the authors, in relation to the comments of the first review, covered all the questions raised. Only one additional suggestion, made by reviewer 2, should be taken into account. As described below:

TOPIC 2.1. Search strategy and eligibility criteria is not clear yet , after text modifications; maybe a table with the word combination strategy used in the present work will be necessary.

Improvement oh this topic is very important for other groups would reproduce the work or use the same estrategy to other studies , this is important for reproducibility of the work and increases the chances of the work being cited.

Sincerely,

José Angelo Lauletta Lindoso

Guest Editor

Fabiano Oliveira

Deputy Editor

The manuscript was revised after some changes made by the authirs. The answers provided by the authors, in relation to the comments of the first review, covered all the questions raised. Only one additional suggestion, made by reviewer 2, should be taken into account. As described below:

TOPIC 2.1. Search strategy and eligibility criteria is not clear yet , after text modifications; maybe a table with the word combination strategy used in the present work will be necessary.

Improvement oh this topic is very important for other groups would reproduce the work or use the same estrategy to other studies , this is important for reproducibility of the work and increases the chances of the work being cited.

Reviewer's Responses to Questions

**Key Review Criteria Required for Acceptance?**

**Methods**

-Are the objectives of the study clearly articulated with a clear testable hypothesis stated?

-Is the study design appropriate to address the stated objectives?

-Is the population clearly described and appropriate for the hypothesis being tested?

-Is the sample size sufficient to ensure adequate power to address the hypothesis being tested?

-Were correct statistical analysis used to support conclusions?

-Are there concerns about ethical or regulatory requirements being met?

Reviewer #1: Objectives, design and analysis all clearly described. Sample size is a function of the relatively sparse number of studies.

Reviewer #2: -Are the objectives of the study clearly articulated with a clear testable hypothesis stated? YES

-Is the study design appropriate to address the stated objectives? NO; BECAUSE THE TOPIC 2.1. Search strategy and eligibility criteria is not clear yet maybe a table for combination word estrategy will be necessary , modified in order to make it clearer 

-Is the population clearly described and appropriate for the hypothesis being tested? YES

Is the sample size sufficient to ensure adequate power to address the hypothesis being tested? NO

-Were correct statistical analysis used to support conclusions? YES

-Are there concerns about ethical or regulatory requirements being met? NO

**Results**

-Does the analysis presented match the analysis plan?

-Are the results clearly and completely presented?

-Are the figures (Tables, Images) of sufficient quality for clarity?

Reviewer #1: Results clearly presented. Good tables and figures.

Reviewer #2: -Does the analysis presented match the analysis plan? YES

-Are the results clearly and completely presented? YES

-Are the figures (Tables, Images) of sufficient quality for clarity? YES

**Conclusions**

-Are the conclusions supported by the data presented?

-Are the limitations of analysis clearly described?

-Do the authors discuss how these data can be helpful to advance our understanding of the topic under study?

-Is public health relevance addressed?

Reviewer #1: Conclusions and limitations all supported.

Reviewer #2: -Are the conclusions supported by the data presented? YES

-Are the limitations of analysis clearly described? YES

-Do the authors discuss how these data can be helpful to advance our understanding of the topic under study? YES

-Is public health relevance addressed? YES

**Editorial and Data Presentation Modifications?**

Reviewer #1: Much Improved manuscript which reads very clearly.

Reviewer #2: "Minor Revision"

TOPIC 2.1. Search strategy and eligibility criteria is not clear yet maybe a table for combination word estrategy will be necessary , modified in order to make it clearer

**Summary and General Comments**

Reviewer #1: A well presented paper

Reviewer #2: the study was made with heterogeneus and low quantity of data but in the other hand the authors get all avialible information to do the work

PLOS authors have the option to publish the peer review history of their article (what does this mean?). If published, this will include your full peer review and any attached files.

Reviewer #1: Yes: Dr Richard Weller

Reviewer #2: Yes: Eduardo Milton Ramos Sanchez

Figure Files:

Data Requirements:

Reproducibility:

References

---

## [Editor Report · Decision Letter 2]

18 Apr 2022

Dear Mr Shita,

We are pleased to inform you that your manuscript 'Prevalence of Leishmania RNA virus in Leishmania parasites in patients with tegumentary leishmaniasis: a systematic review and meta-analysis' has been provisionally accepted for publication in PLOS Neglected Tropical Diseases.

Best regards,

José Angelo Lauletta Lindoso

Guest Editor

Fabiano Oliveira

Deputy Editor

The authors reviewed the manuscripts and accepted the suggestions of reviewer 2. The authors included the word search strategy to carry out this systematic review and meta-analysis. Therefore, I consider that the article should be accepted for publication.

<style type="text/css">p.p1 {margin: 0.0px 0.0px 0.0px 0.0px; line-height: 16.0px; font: 14.0px Arial; color: #323333; -webkit-text-stroke: #323333}span.s1 {font-kerning: none

</style>

---

## [Editor Report · Acceptance letter]

25 May 2022

Dear Mr Shita,

We are delighted to inform you that your manuscript, "Prevalence of Leishmania RNA virus in Leishmania parasites in patients with tegumentary leishmaniasis: a systematic review and meta-analysis," has been formally accepted for publication in PLOS Neglected Tropical Diseases.

Best regards,

Shaden Kamhawi

co-Editor-in-Chief

Paul Brindley

co-Editor-in-Chief
